Corrected: Author correction

# Glacigenic sedimentation pulses triggered post-glacial gas hydrate dissociation

Jens Karstens[1,2], Haflidi Haflidason[1], Lukas W.M. Becker[1], Christian Berndt [2], Lars Rüpke[2], Sverre Planke[3,4], Volker Liebetrau[2], Mark Schmidt[2] & Jürgen Mienert[5]

Large amounts of methane are stored in continental margins as gas hydrates. They are stable under high pressure and low temperature, but react sensitively to environmental changes. Bottom water temperature and sea level changes were considered as main contributors to gas hydrate dynamics after the last glaciation. However, here we show with numerical simulations that pulses of increased sedimentation dominantly controlled hydrate stability during the end of the last glaciation offshore mid-Norway. Sedimentation pulses triggered widespread gas hydrate dissociation and explains the formation of ubiquitous blowout pipes in water depths of 600 to 800 m. Maximum gas hydrate dissociation correlates spatially and temporally with the formation or reactivation of pockmarks, which is constrained by radio-carbon dating of Isorropodon nyeggaensis bivalve shells. Our results highlight that rapid changes of sedimentation can have a strong impact on gas hydrate systems affecting fluid flow and gas seepage activity, slope stability and the carbon cycle.

[1] Department of Earth Science, University of Bergen, Bergen, Norway. [2] GEOMAR Helmholtz Centre for Ocean Research Kiel, Kiel, Germany. [3] Volcanic Basin Petroleum Research (VBPR), Oslo, Norway. [4] The Centre for Earth Evolution and Dynamics (CEED), University of Oslo, Oslo, Norway. [5] CAGE - Centre for Arctic Gas Hydrate, Environment and Climate, UiT - The Arctic University of Norway, Tromsø, Norway. Correspondence and requests for materials should be addressed to J.K. (email: jkarstens@geomar.de)

Continental margin sediments represent the largest methane reservoir on earth[1]. Most methane is stored as free gas or gas hydrate—ice-like clathrates that form under high pressure and low temperature conditions—in the pore space of marine sediments[2]. High gas hydrate saturations can lower the permeability of the host sediment and prevent diffusive migration of gas resulting in its accumulation beneath the base of the gas hydrate stability zone (BGHSZ)[3]. Such gas accumulations result in the build-up of high pore-overpressure, which can lead to the formation or reactivation of focused fluid conduits[4]. Pressurized fluids are released by natural blowout events and create sea floor craters known as pockmarks[5], which are abundant in many gas hydrate provinces around the world. The southern Vøring Plateau gas hydrate province is located at the northern sidewall of the Storegga Slide and received significant amounts of glacigenic sediments related to ice-stream activity during the decay of the Fennoscandian Ice-Sheet at the end of the Last Glacial Maximum[6]. It hosts the Nyegga pockmark field, which consists of several hundred sea floor depressions (Fig. 1, ref. [7]). Seismic data show that gas hydrates occur from beneath the pockmark field towards the South into the Storegga Slide scar[8]. The pockmarks are located on top of pipe structures, which penetrate the gas hydrate layer and root in the free gas zone beneath the BGHSZ or even deeper (Figs. 1b and 2). Several pockmarks host large authigenic carbonate mounds indicating phases with increased and long lasting seepage activity[9].

The dynamic redistribution of gas hydrates within the sediment column due to external forces is a well-established process known as gas hydrate recycling[10,11]. During glacial cycles, gas hydrate dynamics is governed by sea level changes, regional bottom water temperature fluctuations and local sedimentation rate changes. The local sea floor depth is controlled by the global sea level and regional uplift or subsidence. Bottom water temperature changes during deglaciations are mainly controlled by the reorganization of ocean currents and warming of water masses, affecting the sediment temperature profile[12]. This study has a focus on the impact of sedimentation on gas hydrate dynamics. Sediment accumulation influences both the sea floor depth and the sediment temperature profile (Fig. 3). When sediment is deposited, the sediment temperature profile leaves the equilibrium temperature gradient. Conductive heat transfer gradually warms the sedimentary column, which begins with the adjustment of the sediment temperature close to the sea floor and then propagates towards greater depth. As a consequence, gas hydrates at the BGHSZ dissociate into free gas, which dissolves in the pore water or migrate upward, where it may form hydrates again. Only if concentrations are very low, the gas may stay in situ as the buoyancy may not be great enough to overcome capillary forces withholding gas migration. Otherwise the buoyancy of gas and related volumetric expansion of the pore fluid during hydrate dissociation will cause increasing pore overpressure and the formation or reactivation of focused fluid conduits, involving the cracking of sediment formations. This process concentrates gas hydrates at the BGHSZ and reduces sediment permeability, which impedes further gas migration and therefore leads to gas accumulation beneath the BGHSZ (Fig. 3).

We have analysed the gas hydrate dynamics beneath the Nyegga pockmark field with numerical simulations for the period between 30,000 and 15,000 years before present (BP). The analysis built on the global sea level curve[13], local subsidence constraints[14], bottom water temperature information and a local sedimentation history reconstruction. Our dynamic gas hydrate stability simulations integrated time-dependent sediment temperature profiles with a time-dependent sea floor depth reconstruction (Fig. 4a, e). The sea floor depth was calculated combining a global sea level reconstruction[13] with a regional subsidence rate of 1.2 mm/a (ref. [14]) and a local sedimentation rate reconstruction. The sediment temperature profiles used the sedimentation reconstruction as well and were calculated with a finite difference heat-flow simulation building on Fourier's law for heat flow (see "Methods" section). We used a homogenous thermal diffusivity for the glacigenic sediments ($4.2 \times 10^{-7}$ m$^2$/s, ref. [15]). The evolution of the bottom water temperature at the mid-Norwegian margin during the LGM is controversial. The stable oxygen isotope analysis of benthic foraminifera reveals a pronounced δ18O anomaly around the LGM, which may be explained by brine formation[16], freshening by melt water input[17] or warming of immediate water masses[18]. For our simulations, we assume constant bottom water temperature of −1 °C (ref. [15]). We base this assumption on the analysis of stable oxygen isotope

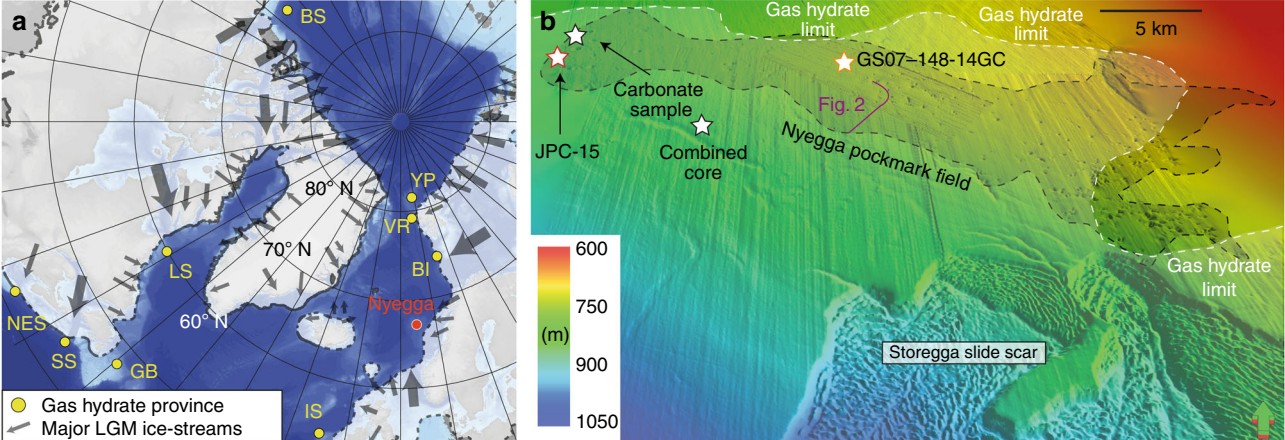

**Fig. 1** Location of the Nyegga pockmark field. **a** Map showing modelled gas hydrate occurrence around the Nordic Sea's shelves, proven gas hydrate deposits (BI Bear Island trough mouth fan and Håkon Mosby mud volcano, BS Beaufort Sea, GB Grand Banks, IS Irish Sea, LS Labrador Sea, NES New England shelf, SS Scotian shelf, VR Vestnesa ridge, YP Yermak Plateau[52], the location of the LGM Fennoscandian, Eurasian and Laurentian ice-sheets[23, 53] and the major ice-streams[54]). The background bathymetry is from GEBCO Digital Atlas published by the British Oceanographic Data Centre on behalf of IOC and IHO (2003). **b** 3D view on seismic bathymetry from the Southern Vøring Plateau showing the location of the Nyegga pockmark field, the extent of gas hydrates[8] (white dashed line) and the Storegga slide scar. Locations of sediments cores, carbonate sample and the combined cores are marked with stars. The bathymetry is from 3D seismic data provided by the Norwegian Petroleum Directorate and extracted using the seismic interpretation software Petrel by Schlumberger

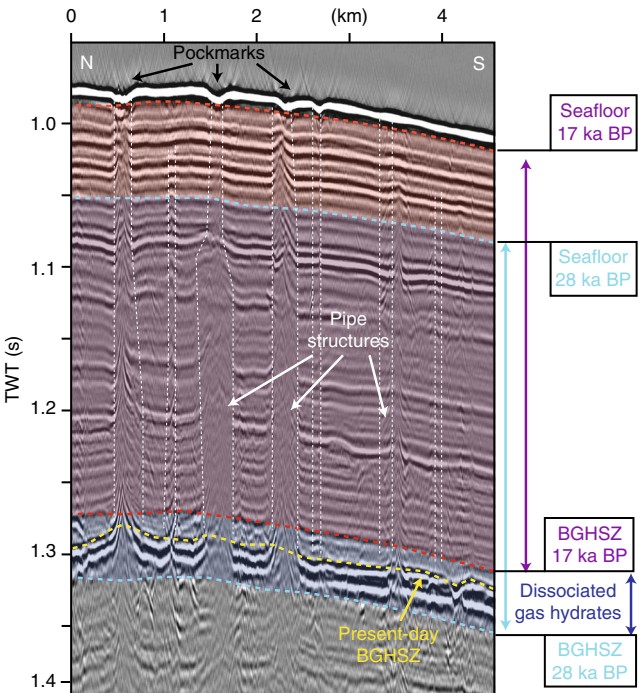

**Fig. 2** Pockmarks and pipe structures at Nyegga. Seismic profile with pipe structures, pockmarks, the present-day base of the gas hydrate stability zone (yellow) and calculated bases of the gas hydrate stability zone at 17 ka BP (blue) and 28 ka BP (red)

from benthic foraminifera from two cores that have been affected by the same bottom water masses. One of the sediment cores from the Vøring Plateau shows stable deep-water conditions during the Last Glacial Maximum[16], while the oxygen isotope record of a core from the northern North Sea indicates that an increase of bottom water temperature did not occur before the end of the Younger Dryas (~11.800 years BP)[19], which correlates with the onset of the Norwegian Atlantic Current[20,21]. All parameters were combined to calculate the pressure and temperature fields (Fig. 4d) to determine the BGHSZ following the approach of Tishchenko et al[22]. We conducted a sensitivity analysis for the input parameters of our simulations by testing different bottom water temperature profiles, sedimentation rate reconstructions, subsidence rates, thermal diffusivity values and geothermal gradients. The sensitivity analysis revealed that variations of these input parameters had no significant impact on the timing and trend of gas hydrate dissociation (see "Methods" secrion and Supplement). The sensitivity analysis includes simulation scenarios, which assume bottom water warming before or during the LGM.

## Results

**Sedimentation rate reconstruction.** The most important input parameter for our simulations is a spatial sedimentation history reconstruction that builds on combining a local age-depth model compiled from 43 radiocarbon dates from five sediment cores (Fig. 4a) into a high-resolution seismo-stratigraphic framework of the study area (see "Methods" section). The sedimentation rate reconstruction shows distinct differences between periods characterized by decreased sedimentation rates between 0.5 and 1 mm/a, and phases of increased rates of glacigenic input with peak rates of 9 mm/a (Fig. 4b). The sedimentation rate shows a first maximum at 28,000 years BP, which coincides with the first advance of the Fennoscandian ice sheet on the nearby shelves[23]. At around 25,000 years BP, sedimentation rates started to

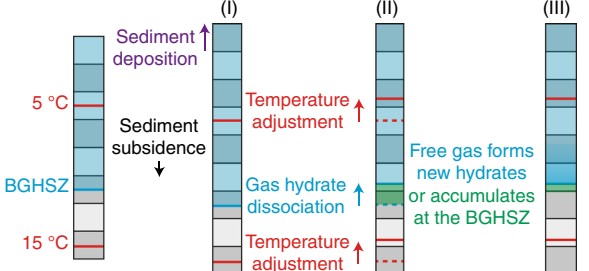

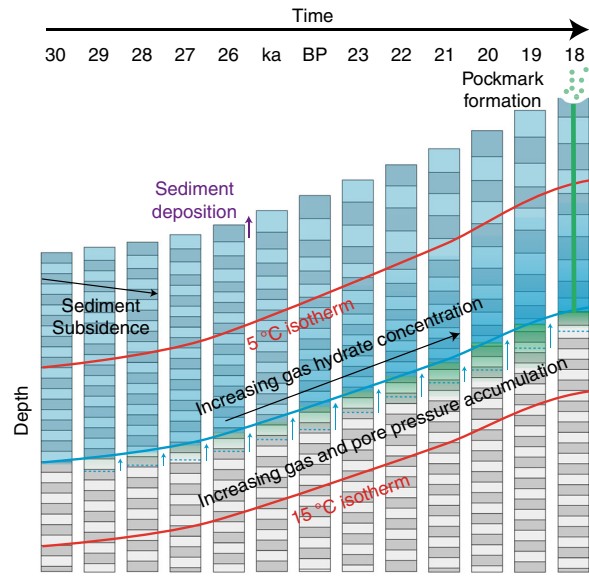

**Fig. 3** Processes influencing the gas hydrate dynamics within marine sediments. **a** Sediment gets deposited and the sedimentary column subsides (I), the temperature profile and the base of the gas hydrate stability zone (BGHSZ) adjusts and gas hydrate dissociates (II). Free gas migrates back in the GHSZ and forms hydrates again, while hydrate concentration increases and free gas accumulates beneath the base of the GHSZ (III). **b** Conceptual evolution of isotherms, hydrate concentrations and free gas accumulations during ongoing sedimentation and subsidence for a single location

increase again and had a second maximum at 20,500 years BP, which correlates with the deposition of glacial debris lobes. There was a third maximum between 19,500 and 18,500 years BP associated with sediment-rich melt water plumes[6].

**Modelling the dynamic changes of the hydrate stability zone.** The finite difference heat flow simulations demonstrate, how the diffusive heat transfer warms the newly deposited sediments and affects the sediment temperature profile and the BGHSZ (Figs. 3 and 4d). The sediment temperature adjustment causes a significantly shoaling of the BGHSZ within the sedimentary column during the simulation period (Fig. 4d). The change of the BGHSZ is the result of the combined effects of sedimentation, sea level change and subsidence. In the period between 30,000 and 15,000 years BP, the eustatic sea level fluctuation alone would have resulted in a maximum BGHSZ shoaling of ~10 m, while continuous local subsidence would have caused a deepening of the BGHSZ of ~3 m over the entire simulation period (Fig. 4e). During the same period, the sedimentation alone would have

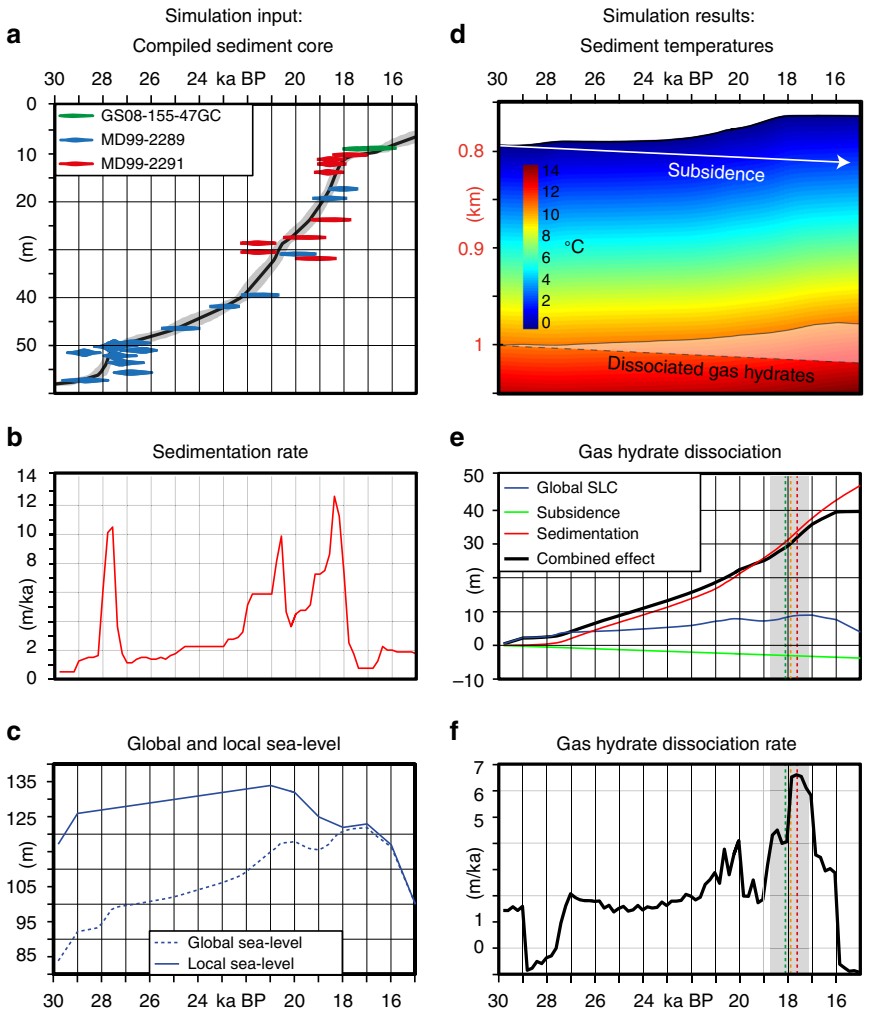

**Fig. 4** Numerical simulation of the gas hydrate stability zone. **a** Age-depth-model and radiocarbon ages from the combined core. **b** Sedimentation rate evolution. **c** Global and local sea level anomaly in comparison to the present-day sea level and phases with enhanced methane seepage. **d** Sediment temperature reconstruction and the effect on gas hydrate dissociation (compare with Fig. 2b). **e** Absolute gas hydrate dissociation with time: effect of sedimentation (red line), effect of subsidence (green line), effect of sea level fluctuation (blue line), effect of all combined (black line) and phase with enhanced methane seepage (green dashed line mean calibrated age of carbonate sample, orange dashed line mean calibrated age of Isorropodon nyeggaensis horizon in core GS07-148-14GC, red dashed line mean calibrated age of Isorropodon nyeggaensis horizon in core JCP-15 (ref. [54]) and grey box, indicating the error margins of the calibrations). **f** Gas hydrate dissociation rate and phase with enhanced methane seepage

caused a shoaling of the GHSZ by ~47 m outpacing the effects of sea level fluctuation and subsidence by far. All factors combined resulted in a net dissociation of ~40 m. Therefore, sedimentation was the most important driver of gas hydrate dynamics in the study area during and after the Last Glacial Maximum (Fig. 4f). A possible bottom water temperature increase after the Last Glacial Maximum would have amplified further sedimentation-related gas hydrate dissociation. The gas hydrate dissociation rate follows the trend of sedimentation rate with a delay of about 1000 years having its maximum around 17,800 years BP.

**Timing of enhanced seepage from radiocarbon analysis.** High-resolution 3D seismic analysis of the Nyegga fluid pipes indicates that sea floor seepage is episodic, correlating with Pleistocene climate fluctuations[24]. Reflection seismic data and radiocarbon dating of sediment cores reveal that the pipes crosscut sediments, which was deposited before and during the LGM (Fig. 2). This is a clear indicator for seepage activity during at the end of the LGM. To have more detailed constraints about the timing of enhanced methane seepage during that period, we further

analysed seep fauna and seafloor carbonates. The chemosynthetic symbiotic bacteria containing bivalve species *Isorropodon nyeggaensis* (Fig. 5; ref. [25]) provide a model-independent time marker for methane seepage in the Nyegga pockmark field. Sediment coring (GS07-155-14GC in Fig. 1b) of a seep-related mound structure with a central depression revealed a 60 cm-thick layer of *Isorropodon nyeggaensis* shells at a depth of 100 cm (Fig. 5). Radiocarbon dating of these shells points towards two episodes of pronounced methane seepage about 17,900 and about 15,700 years BP[26]. A similar shell-bearing horizon was cored within a pockmark (JPC-15) in the western part of the field, providing analogue ages of around 17,600 years BP[26]. Additionally, mineralogical and geochemical analyses of a sea floor-exposed sediment crust (Fig. 1b) indicate a maximum age around 18,000 years BP (see "Methods" section). The sample consists of 30–50 wt.% of Ca(Mg)-carbonate with calcite and dolomite in approximately equal abundancy, which is similar to sediment samples from JPC-15 (ref. [27]). The bulk $\delta^{13}C$ values between −0.4 and −5.2‰ relative to Vienna Pee Dee Belemnite for binary mixtures of calcite and dolomite from three sub-samples suggest

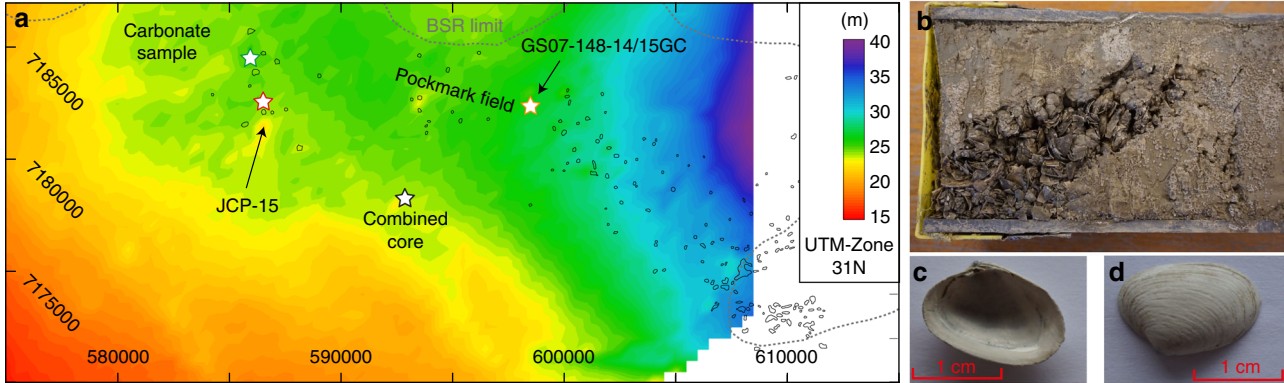

**Fig. 5** Simulated distribution of gas hydrate dissociation. **a** Map showing the distribution of gas hydrate dissociation at 17.8 ka with pockmark location[7], extent of gas hydrates indicated by a BSR (Bottom-simulating reflector; dashed black lines[8]), and coring and sample locations. **b** Photograph showing the beginning of section 2 of gravity core GS07-148-14GC with a pronounced accumulation of intact and fragmented shells of Isorropodon nyeggaensis. **c**, **d** Photographs of Isorropodon nyeggaensis shell from this core

an endmember $\delta^{13}C$ signature of −11‰ for the dolomitic mineral phase. This value is less negative than known from other authigenic carbonate concretions from the study area[28], however, still indicative for seep-related overgrowth or recrystallization of biotic mineral precursors[29,30]. Probably the dolomitic phase precipitated either outside the main methane seep site from pore fluids ascending with the methane or after vigorous methane seepage ceased. The $\delta^{18}O$-values of the carbonaceous mixtures indicate a slight enrichment of $^{16}O$ in dolomite compared to calcite. This is contrary to the oxygen fractionation during precipitation at the same sea floor temperature and possibly rather reflects pore water freshening than increased temperatures during dolomite formation[31]. These observations combined, suggest a simultaneous pulse of methane seepage in the pockmark field starting around 18,000 years BP.

**Excess pore pressure quantification**. The overpressure resulting from the buoyancy of free gas can be calculated by $P = H_{gas} \times g \times (\rho_w - \rho_{CH4\_10\ MPa})$[32], where $H_{gas}$ represents the gas column height, $g$ is gravitational acceleration (9.81 m/s²), $\rho_w$ is density of the formation water (~1025 kg/m³) and $\rho_{CH4\_10\ MPa}$ is density of methane for pressure of 10 MPa C (~84 kg/m³; ref. [33]). The volume of the gas formed by gas hydrate dissociation is 164 times greater than the hydrate volume under atmospheric pressure conditions[34], while density of methane under atmospheric pressure condition (0.656 kg/m³) is about 128 times smaller than at 10 MPa. The gas column height is consequently: $H_{gas} = 164/128\ H_{hydrate} \times s_{hydrate}$, where $s_{hydrate}$ is the gas hydrate concentration (3–12%, ref. [35]) and $H_{hydrate}$ is the height of dissociated gas hydrates (40 m from simulations). The resulting overpressure due to buoyancy lies between 15.1 and 60.6 kPa.

## Discussion

The radiometrically-deduced timing of enhanced seepage coincides well with the independently modelled maximum of gas hydrate dissociation around 17,800 years BP (Fig. 4). For this time slice, the gas hydrate dissociation peaks (Fig. 4h) and the simulation even reproduces the location of the western pockmark group (Fig. 5). These areas received most of the gas released by dissociation due to the locally increased gas hydrate dissociation rate and as the result of the topography of the BGHSZ focussing lateral gas migration towards the pockmark field[8]. The temporal and spatial correlation between the maximum gas hydrate dissociation and the pockmark activity strongly suggests that gas hydrate dynamics related to increased glacigenic sediment

accumulation controlled methane seepage after the Last Glacial Maximum at the Nyegga pockmark field.

Gas hydrate dissociation results in high pore overpressure by volume expansion during phase transition and buoyancy of the released gas. Considering 40 m of dissociated hydrates and hydrate concentrations of 3–12%[35,36], the buoyancy of the resulting gas column would cause pore-overpressures of 15–61 kPa. This calculation assumes that gas formed by the gas hydrate dissociation is mobile and accumulates beneath the BGHSZ, which is plausible considering the well-developed bottom-simulating reflector, indicating the presence of a gas column beneath the highly dynamic BGHSZ (Fig. 3).

More detailed quantification of excess pore-pressure due to volume expansion would require detailed knowledge of past gas hydrate concentration, distribution and hydraulic properties of the host sediments. However, excess pore pressure is expected to be highest at times of maximum gas hydrate dissociation rate[37] and hence to correlate with the timing of enhanced methane seepage.

Depending on the average gas hydrate concentration, the methane seepage potential from gas hydrate dissociation in the modelled area accrues to between 26 and 212 Mt (see "Methods" section). Our model covers only a tenth of the local gas hydrate province, implying a total seepage potential of >1 Gt of methane. Ice-stream activity and the rapid melting of the Fennoscandian, Eurasian and Laurentide ice-sheets released pulses of high sedimentation discharge to gas hydrate provinces along the North Atlantic and Artic Ocean margins (Fig. 1a). This high sediment discharge might have triggered widespread and simultaneous gas hydrates dissociation, implying a global seepage potential of several billion tons of methane. Even if a large fraction of the methane has been recycled within the hydrate stability zone or consumed by the benthic filter, it is likely that the sudden hydrate dissociation due to sediment input has released large amounts of methane into the water column. It is difficult to constrain, if the methane release from pockmarks at Nyegga occurred continuously with low seepage rates or catastrophically with high seepage rates comparable to drilling-induced blowout events. Numerical simulations indicate that methane released from natural seeps deeper than 100 m below sea level will not reach the atmosphere via bubble transport due to oxidisation and dissolution, while a catastrophic methane release allows a more efficient transport of methane to the sea surface[38]. Consequently, it is speculative if the focused methane release at Nyegga after the LGM directly affected the atmosphere methane budget. Nevertheless, the dissolution and oxidisation of methane in the water column has likely reduced the ocean's potential of absorbing

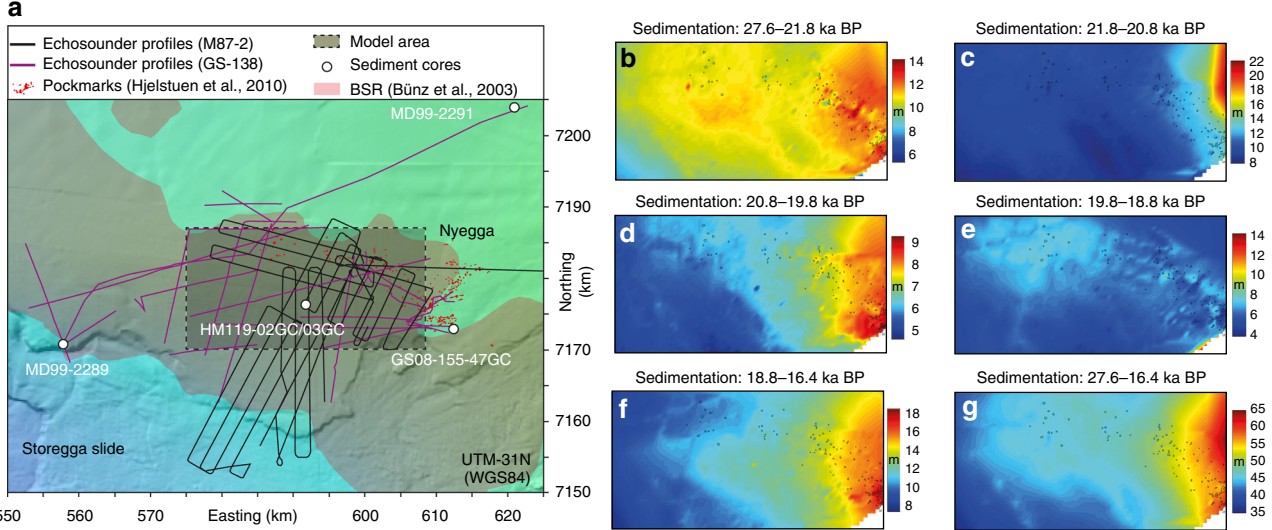

**Fig. 6** Reconstruction of the seismo-stratigraphic framework of the Nyegga area. **a** Map showing the location of the echosounder profiles used for the seismo-stratigraphic framework, pockmarks, sediment core locations, model area, and area with mapped BSR (Bottom-simulating reflector). The map was created with IHS Kingdom Suite and data sourced from bathymetry grids created by Norsk Hydro AS[55]. **b–g** Sediment thickness maps for different intervals of the seismo-stratigraphic framework

atmospheric methane. Assuming that the sedimentation controlled gas hydrate dissociation and methane seepage occurred simultaneously around glaciated gas hydrate provinces after the LGM, this process would have contributed to the increase of global atmospheric methane concentrations after the Last Glacial Maximum[39].

Our simulations indicate that sedimentation on its own is able to cause significant hydrate dissociation independent of bottom water temperature warming or sea level fluctuation. The link between sedimentation, gas hydrate dynamics and pore-pressure evolution is a key to understand slope stability. Glacial and fluvial depocenters on passive margins host some of the largest submarine slope failures. The resulting landslides have mainly been attributed to pore-overpressure due to increased sediment load or gas hydrate dissociation, which is commonly explained by post-glacial bottom water temperature warming[40]. However, many large slope failures occurred after phases with enhanced sediment accumulation with a delay of several thousand years[41]. We observe a similar delay for hydrate dissociation, in which the timing is mainly controlled by thermal sediment properties. Although there is so far no proof for hydrate-related slope failure, our results show that sedimentation-induced gas hydrate dissociation may play an important part in pore-overpressure accumulation and trigger or precondition slope failures in trough mouth and river fan systems.

Unlike sea level and bottom water temperature fluctuations, the effect of sedimentation on the absolute thickness of the gas hydrate stability zone is relatively small, because basin subsidence counteracts sedimentation and heat flow re-adjusts sediment temperatures. However, sedimentation results in a large shift of the BGHSZ within the sedimentary column and causes the redistribution of large volumes of hydrates (Figs. 3b and 4f). If redistribution is not possible via diffusive fluid flow due to lithological or gas hydrate-related permeability barriers, it has the potential to create pore-overpressure and trigger focused fluid flow. The discharge of methane via focused fluid flow has significant impact on local and global carbon budgets and fluxes[42]. Therefore, sedimentation has to be considered, when understanding gas hydrate and fluid flow systems not only in glaciated margins, but also in other areas with high sedimentation rates. Increased sedimentation is key to understand the formation of

focused fluid conduits and slope failures in gas hydrate systems around the world.

## Methods

**Seismo-stratigraphic framework**. The seismo-stratigraphy of the Nyegga area is characterized by the presence of six pronounced reflections, which are well-defined in echosounder data throughout the study area[43]. We traced these reflection in a dense network of echosounder profiles collected with the TOPAS echosounder system on-board RV G.O. Sars by University in Bergen in 2004 and a Parasound echosounder system on board of RV Meteor by Geomar in 2012 (Fig. 6). The stratigraphic section defined by the deepest and shallowest interpreted seismic reflections consists of spatially homogenous, laminated sediments and shows no unconformities indicating phases of erosion. The thickness of specific intervals within this stratigraphic section varies depending on the proximity to glacigenic depositional centres to the West[7]. The seismo-stratigraphy represents a continuous record of the sedimentation history of the study area and allows constraining a stratigraphic framework for the Nyegga pockmark field.

**Radiocarbon age calibration**. The calibration of radiocarbon dates from marine sources need to be corrected for the global and local reservoir effects. The global reservoir effect is set to 405 years within the Marine13 calibration curve[44]. The local reservoir effect depends on time-varying oceanographic parameters, which are not well-constrained for the study area during the analysed period. Numerical modelling of the local reservoir effect for the North Atlantic suggests values between 200 and 600 years[45], while the comparison of terrestrial and marine $^{14}$C reservoir ages revealed variations of the local reservoir effects between 100 and 400 years for the Younger Dryas[46]. To acknowledge this uncertainty, we used a local reservoir effect of 400 years with an uncertainty of ± 200 years for the calibration tool Calib7.1 using the Marine13 calibration curve for all radiocarbon based dates in this study[43,47] to make them self-consistent.

**Age-depth-model**. The age-depth model builds on integrating radiocarbon dates from the five sediment cores MD99-2289, MD99-2291, HM119-02GC, HM119-03GC and GS08-155-47GC (Fig. 7; Table 1). Sediment core MD99-2291 is located close to the depositional centre in the West of the study area and has a high-resolution record of the shallow to intermediate interval of the seismo-stratigraphic framework, while sediment core MD99-2289 is located more than 60 km away close to the Storegga Slide side-wall and provides age information for the intermediate to deep interval of the seismo-stratigraphic framework. Sediments cores HM119-02GC, HM119-03GC and GS08-155-47GC targeted the shallowest seismic reflections of the seismo-stratigraphic framework.

The analysis of the seismo-stratigraphy of the modelled area suggests that the shape of the sedimentation curve is approximately constant throughout the study area and only the magnitude of sedimentation varies depending on the proximity to the depositional centres. This implies that the relative depth of a specific date within an interval of the stratigraphic framework at its coring location should be approximately constant within the stratigraphic framework. Therefore, we transferred the depth from

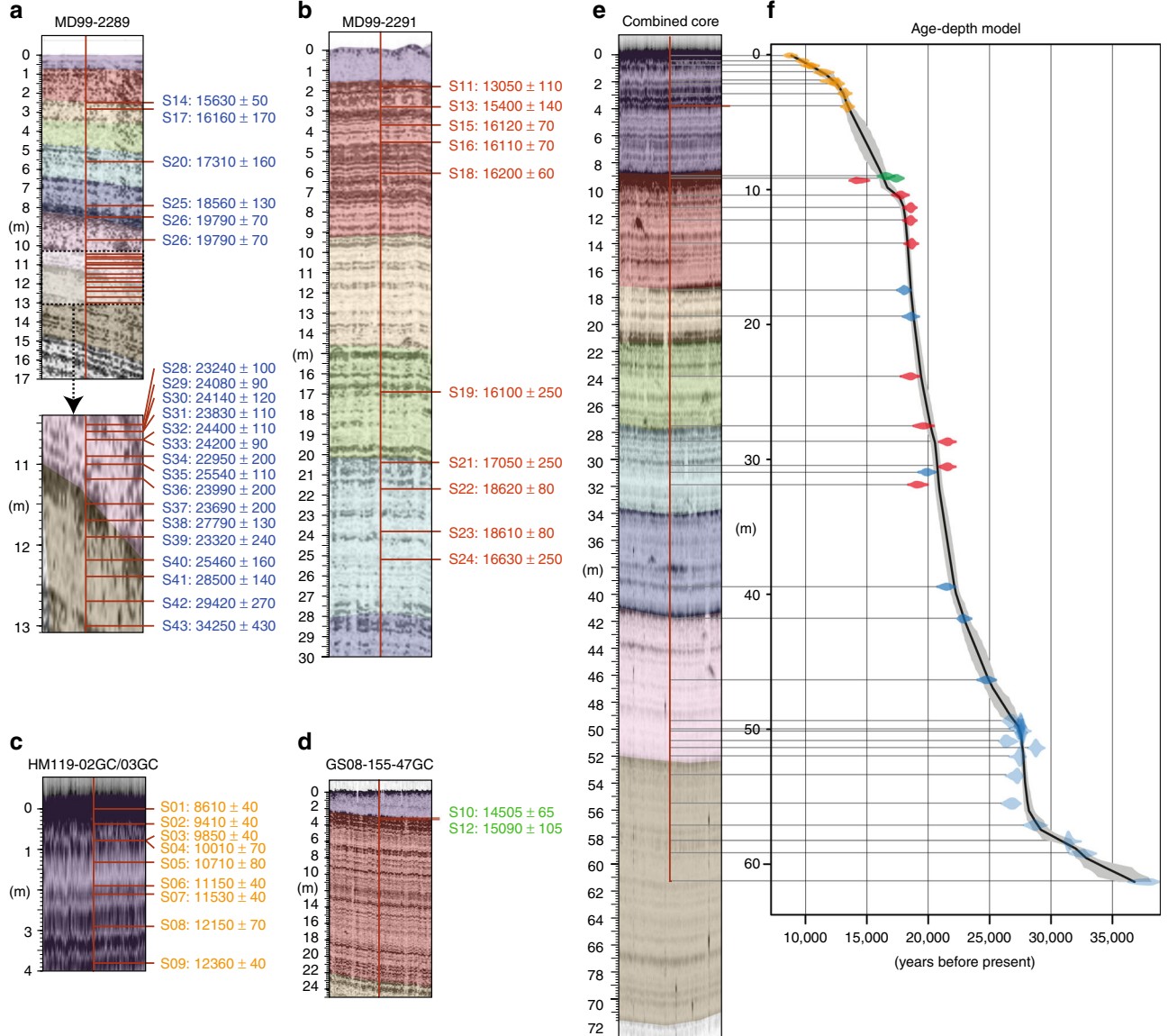

**Fig. 7** Sedimentation rate reconstruction based on combined sediment cores from the Nyegga area. **a–d** Echosounder profiles showing the locations of radiocarbon dates from the different sediment cores. The intervals of the seismo-stratigraphic framework are coloured. **e** Locations of the radiocarbon dates in the combined core. **f** Age-depth model from the combined core (calibrated ages) with the 2 sigma standard deviation in grey

cores MD99-2289, MD99-2291, HM119-03GC and GS08-155-47GC, to the coring location of HM119-03GC with the following equation:

$$\text{depth}_k^0 = \text{top}_i^0 + \frac{\text{top}_i^j - \text{depth}_k^j}{\text{base}_i^j - \text{top}_i^j} \times (\text{base}_i^0 - \text{top}_i^0),$$

whereas $\text{depth}_k^0$ is the transferred depth of sample k from core site j, which lies in coring depth $\text{depth}_k^j$ within the stratigraphic interval i. The radiocarbon ages, including their measurement errors and their transferred depth within the virtual core were the input for the Bayesian statistics based age-depth modelling script Bacon[48], which provided a high-resolution age-depth model for the Nyegga pockmark field (Fig. 3d; Fig. 7). The cores MD99-2289, MD99-2291 and GS08-155-47GC provide overlapping age-depth information for the section between 8 and 33 m of the combined core (7 f). The age-depth information from MD99-2289 and MD99-2291 reveal very similar sedimentation rates. The age-depth information provided by GS08-155-47GC is in good agreement with the trend from MD99-2289 and MD99-2291. The shallow-most radiocarbon age from MD99-2291 is about 1500 years younger than the shallow-most radiocarbon age from GS08-155-47GC. However, the Bayesian age-depth modelling prefers the age-depth information from GS08-155-47GC leading to a smoother sedimentation curve than following the shallow-most radiocarbon age from MD99-2291. The "too young" radiocarbon age of the shallow-most radiocarbon age from MD99-2291 is most likely the result of reworking as the sample was taken directly below an erosional surface (between purple and red intervals in 7b).

**4D sedimentation reconstruction**. The local age-depth model and the seismo-stratigraphic framework were then integrated into 4D sedimentation history reconstruction. The age-depth-model provided absolute age information for each seismic reflections and the local sedimentation rate evolution. The sedimentation rate constrained by the age-depth model was then adjusted for every location within the seismo-stratigraphic framework by scaling the sedimentation rate for each interval and each location of the seismo-stratigraphic framework with the thickness of that interval at a specific position divided by the thickness of the interval in the combined core. This approach was not possible for the stratigraphic record younger than 16.2 ka BP and for the western-most part of the Nyegga pockmark field, where buried glacigenic debris lobes were deposited in between the laminated sediments.

**Analyses of carbonate crusts sampled from Nyegga pockmark area**. A carbonate crust sample was collected during a dive on submersible MIR (on board RV Akademik Mstislav Keldysh during GEOMAR cruise 40 in 1998). The sample was analysed for C and O isotope composition by dissolving three finely powdered subsamples with 100% $H_3PO_4$ at 72 °C in the Carbo Kiel IV preparation line and thereby releasing $CO_2$, which was isotopically analysed using a Finnigan MAT 253 mass spectrometer. $^{18}O/^{16}O$ and $^{13}C/^{12}C$ ratios were determined against a working calcite standard and are expressed in the common δ-notation relative to Vienna Pee Dee Belemnite (VPDB). The uncertainty of this sample set accompanying standard measurements ($n = 8$) was 0.16‰ (2σ) for $\delta^{18}O$ and 0.04‰ (2σ) for $\delta^{13}C$.

**Table 1 Radiocarbon dated samples used for the age-depth model and sedimentation history reconstruction. Radiocarbon ages are not corrected for reservoir effect**

| ID | Sample name | Core name | Sample depth (cm) | Combined core depth (cm) | Radiocarbon age (a) | St. dv. (a) | Reference |
|---|---|---|---|---|---|---|---|
| S01 | Beta-226957 | HM119-02GC | 6.5 | 5 | 8610 | 40 | This study |
| S02 | Beta-226958 | HM119-02GC | 54 | 45 | 9410 | 40 | This study |
| S03 | Beta-226959 | HM119-02GC | 95 | 80 | 9850 | 40 | This study |
| S04 | Beta-229428 | HM119-03GC | 13.25 | 83 | 10010 | 70 | This study |
| S05 | Beta-229429 | HM119-02GC | 155.5 | 131 | 10710 | 80 | This study |
| S06 | Beta-226960 | HM119-03GC | 146 | 195 | 11150 | 40 | This study |
| S07 | Beta-226961 | HM119-03GC | 170.25 | 215 | 11530 | 40 | This study |
| S08 | Beta-229427 | HM119-03GC | 256.5 | 288 | 12150 | 70 | This study |
| S09 | Beta-226962 | HM119-03GC | 372 | 385 | 12360 | 40 | This study |
| S10 | ETH-38626 | GS08-155-47GC | 319 | 900 | 14505 | 65 | This study |
| S11 | ETH-25495 | MD99-2291 | 180 | 932 | 13050 | 110 | [6] |
| S12 | ETH-38628 | GS08-155-47GC | 339 | 916 | 15090 | 105 | This study |
| S13 | ETH-25958 | MD99-2291 | 280 | 1040 | 15400 | 140 | [6] |
| S14 | Beta-380040 | MD99-2289 | 249 | 1745 | 15630 | 50 | [57] |
| S15 | Poz-3950 | MD99-2291 | 367 | 1134 | 16120 | 70 | [6] |
| S16 | Poz-3951 | MD99-2291 | 455 | 1229 | 16110 | 70 | [6] |
| S17 | AAR-6235 | MD99-2289 | 285 | 1941 | 16160 | 170 | [43] |
| S18 | Poz-3952 | MD99-2291 | 614.5 | 1401 | 16200 | 60 | [6] |
| S19 | ETH-22959 | MD99-2291 | 1690 | 2384 | 16100 | 250 | [6] |
| S20 | AAR-6236 | MD99-2289 | 563 | 3094 | 17310 | 160 | [43] |
| S21 | ETH-22960 | MD99-2291 | 2040 | 2751 | 17050 | 250 | [6] |
| S22 | Poz-3956 | MD99-2291 | 2168 | 2869 | 18620 | 80 | [6] |
| S23 | Poz-3957 | MD99-2291 | 2375 | 3055 | 18610 | 80 | [6] |
| S24 | ETH-22961 | MD99-2291 | 2523 | 3187 | 16630 | 250 | [6] |
| S25 | ETH-25496 | MD99-2289 | 790 | 3944 | 18560 | 130 | [43] |
| S26 | Beta-380041 | MD99-2289 | 850 | 4180 | 19790 | 70 | [57] |
| S27 | ETH-25497 | MD99-2289 | 970 | 4635 | 21370 | 160 | [43] |
| S28 | Beta-376421 | MD99-2289 | 1050 | 4938 | 23240 | 100 | [57] |
| S29 | Beta-376422 | MD99-2289 | 1060 | 4976 | 24080 | 90 | [57] |
| S30 | Beta-373227 | MD99-2289 | 1063 | 4987 | 24140 | 120 | [57] |
| S31 | Beta-365943 | MD99-2289 | 1065 | 4995 | 23830 | 110 | [57] |
| S32 | Beta-376423 | MD99-2289 | 1070 | 5014 | 24400 | 110 | [57] |
| S33 | Beta-380041 | MD99-2289 | 1073 | 5025 | 24200 | 90 | [57] |
| S34 | ETH-23313 | MD99-2289 | 1090 | 5090 | 22950 | 200 | [43] |
| S35 | Beta-380043 | MD99-2289 | 1103 | 5139 | 25540 | 110 | [57] |
| S36 | ETH-25498 | MD99-2289 | 1120 | 5204 | 23990 | 200 | [43] |
| S37 | ETH-23314 | MD99-2289 | 1150 | 5346 | 23690 | 200 | [43] |
| S38 | Beta-380044 | MD99-2289 | 1172 | 5459 | 27790 | 130 | [57] |
| S39 | ETH-25499 | MD99-2289 | 1189 | 5551 | 23320 | 240 | [43] |
| S40 | ETH-23315 | MD99-2289 | 1220 | 5711 | 25460 | 320 | [43] |
| S41 | Beta-380045 | MD99-2289 | 1243 | 5832 | 28500 | 140 | [57] |
| S42 | ETH-24870 | MD99-2289 | 1260 | 5921 | 29420 | 270 | [43] |
| S43 | ETH-24871 | MD99-2289 | 1300 | 6132 | 34250 | 430 | [43] |

**Table 2 Mineralogical and geochemical characterisation of carbonate concretion**

| ID | Dissolvable fraction[a] | Calcite/dolomite ratio[b] | $\delta^{13}C_{bulk}$ (‰ VPDB) | 2SD (‰ VPDB) | $\delta^{18}O_{bulk}$ (‰ VPDB) | 2SD (‰ VPDB) |
|---|---|---|---|---|---|---|
| | (wt.%) | | | | | |
| Mie1-3729-1 | 33 | 1.32 | -0.44 | 0.04 | 0.91 | 0.18 |
| Mie1-3729-2 | 46 | 0.46 | -5.22 | 0.07 | -0.67 | 0.07 |
| Mie1-3729-3 | 38 | 0.54 | -3.84 | 0.08 | -0.74 | 0.10 |

[a]dissolved by 2.25 N HNO3
[b]XRD detection

Table 2 shows the results with their individual measurement uncertainties on 2 SD level. No further corrections for different acid fractionation factors of calcite and dolomite were applied. Mineral composition of fine-grained carbonaceous material was determined by using X-ray diffraction analysis (Philips PW 1820, Co-Kα, 0.02°/s). X-ray diffraction patterns were interpreted by using the XPowder® software. Identified carbonate minerals (i.e. calcite and dolomite) were quantified by Rietveld refinement. In addition, we applied U–Th geochronology methods for methane-derived authigenic carbonates to constrain the precipitation age of the carbonates[49,50]. However, the analysis indicated different formation ages for the calcitic

and dolomitic dominated subsamples of the same carbonate sample. A too high contribution of detrital Th in both carbonate phases prevented deducing reasonable isochron-based age information. The finding of different formation ages for the different carbonate phases is in agreement with the light stable isotope findings. The $^{14}C$ data from the carbonate crust sample Mie 1-3729-1 represent a rather robust maximum formation age of 18,000 years BP (15,710 14 C). This maximum age interpretation is based on the assumption that any carbon contributed from the sediment system below would be relatively depleted in $^{14}C$ by ongoing decay during burial, resulting in apparent higher ages.

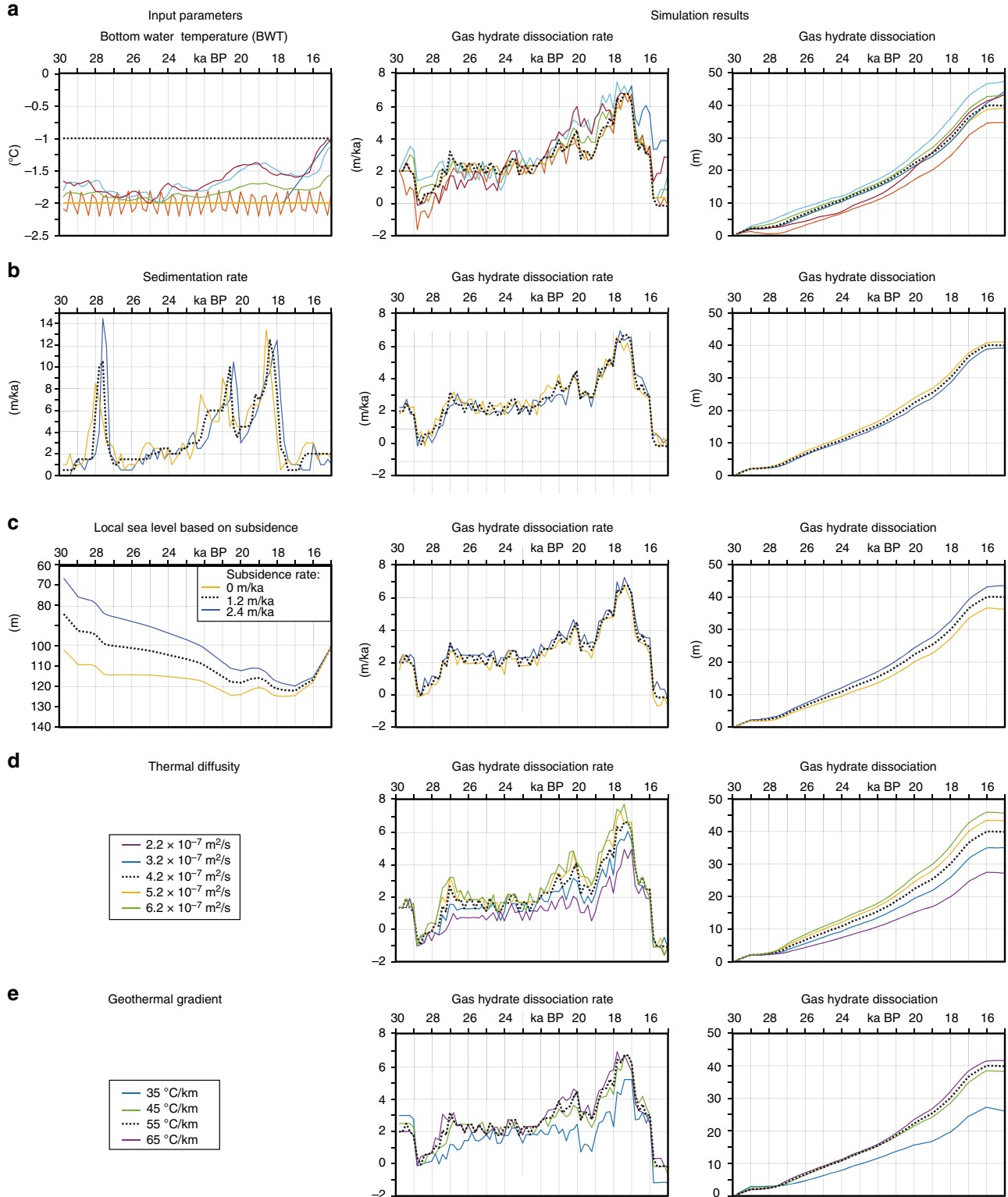

**Fig. 8** Results of the sensitivity analyses for different simulation input parameters. **a** Bottom water temperature profiles and the resulting gas hydrate dissociation rates and absolute gas hydrate dissociation curves. **b** Sedimentation rates: minimum, maximum and best-fit sedimentation rate reconstructions and the resulting gas hydrate dissociation rates and absolute gas hydrate dissociation curves. **c** Subsidence rate as well as their impact on the local sea level anomaly and the resulting gas hydrate dissociation rates and absolute gas hydrate dissociation curves. **d** Thermal diffusivity and the resulting gas hydrate dissociation rates and absolute gas hydrate dissociation curves. **e** Geothermal gradient and the resulting gas hydrate dissociation rates and absolute gas hydrate dissociation curves[56]

**Finite difference temperature modelling**. The temperature profile modelling builds on Fourier's law for heat flow and the assuming homogenous thermal properties of the deposited glacigenic sediments, which allows using the thermal diffusivity $\kappa$ ($4.2 \times 10^{-7}$ m$^2$/s; ref. [15]) for calculating the heat flow: $q = \kappa \nabla T$. We transferred this simple differential equation in a 1D finite difference algorithm:

$$T_i^{n+1} = \frac{\kappa \Delta t}{\Delta x^2}\left(T_{i+1}^n - 2T_i^n + T_{i-1}^n\right) + T_i^n,$$

whereas $\Delta t$ is the temporal discretization of 1 s and $\Delta$s the spatial discretization of 0.1 meter. The simulation started with equilibrium temperature profile with a length of 300 m following the local temperature gradient of 55 °C/km (ref. [15]). This profile was then extended at the top by the amount of sediments deposited within a time step of 200 years according to the 4D sedimentation history reconstruction. The sediments had the bottom water temperature of −1 °C, which was also used as boundary condition, the lower boundary was defined as a gradient and the heat flow adjustment was simulated over 200 years using an implicit approximation. The resulting temperature profile was then the starting model for the following time step.

**Gas hydrate stability calculation**. The gas hydrate stability calculations followed the approach by Tishchenko[22], which build on thermodynamic stability and solubility calculation for methane hydrates. The dissociation temperature for specific depths below sea floor z is calculated following the empirical function: $T_{diss}(z) = a + b \times \log(P(z)) + c \times (d − \log(P(z)))^3$, whereas $T_{diss}$ is the dissociation temperature, $P$ is the hydrostatic pressure and $a$, $b$, $c$, $d$ are empiric parameters calculated by fitting the stability curve of methane hydrates for a given salinity of 3.5% and pure methane ($a = −25.1391$, $b = 7.9832$, $c = −0.0959$ and $d = 6.0687$). The hydrostatic pressure at the sea floor and within the top 300 m in the sediments for a given point at a given time was calculated analytically by implementing the sea level fluctuation, subsidence and the sedimentation reconstruction. The $T_{diss}$ profile was then compared with the temperature profile from the finite difference heat flow simulations to determine the BGHSZ, which is defined by the shallowest depth in the modelled temperature is higher than $T_{diss}$.

**Sensitivity analysis**. In order to test the robustness of our simulations, we performed sensitivity analyses for different sedimentation rates, bottom water temperature profiles, the subsidence rates affecting the local sea level curve, bulk thermal conductivity values and the geothermal gradients (Fig. 8; the input parameter and simulations results used in the main part of the manuscript are marked with dashed black lines). We tested the impact of bottom water temperature on the gas hydrate dynamics by applying different temperature evolution profiles. These simulations included scenarios with constant temperatures of −1 °C (dashed black line in Figs. 8a) and −2 °C (yellow line) and repeatedly fluctuating temperatures between −1.75 and −2.25 °C (orange line). These input parameters result in very similar curves for the dissociation rate of gas hydrates. In addition, we performed simulations with four potential warming scenarios with an absolute warming of ~0.5 °C (green line) and ~1 °C (red, light blue and dark blue). All simulations result in similar shaped gas hydrate dissociation curves with a peak of dissociation after 18,000 years before present. The absolute thickness of gas hydrate dissociation at the end of the simulations lies between 35 and 47 m, which shows that bottom water temperatures had only a secondary impact on gas hydrate dynamic during the LGM at Nyegga. The solution space of the Bayesian age-depth reconstruction allows different sedimentation rate reconstructions (Fig. 8b). In addition to the statistically most likely reconstruction (dashed black line), we tested the sedimentation rate reconstructions, which bound the solution space (blue and yellow lines; marked in grey in Fig. 7f). These sedimentation rate reconstructions result in very similar gas hydrate dissociation rates and absolute dissociation curves. The analysis of different local subsidence rate (Fig. 8c), which influence the local sea level curve reveals that subsidence had only a minor impact on gas hydrate dynamics, even when neglecting subsidence (yellow line) or assuming a twice as high subsidence rate (blue line). The analyses of the thermal diffusivity of the sediments and the geothermal gradients reveal that the effect of sedimentation on gas high dynamics becomes more important for sediments with a high thermal diffusivity and in areas with high geothermal gradients (Fig. 8d, e). However, the timing of enhanced gas hydrate dissociation remains stable for various thermal property values. The sensitivity analysis revealed that the gas hydrate dissociation rate and the absolute value of dissociated gas hydrates are robust and the observed gas hydrate dynamics can be observed using a wide range of parameters.

**Seepage potential calculation**. The seepage potential can be calculated by multiplying the average thickness of dissociated gas hydrates (40 m from our simulations) with the area of gas hydrate occurrence A (model area: 540 km$^2$), the gas hydrate concentration (3–12%, ref. [35]), the sediment porosity $\Phi$ (0.38–0.76, ref. [51]), the gas hydrate expansion factor under atmospheric pressure conditions (164, ref. [34]) and the density of methane under atmospheric conditions $\rho_{CH4\_atmospheric}$ (~ 0.656 kg/m$^3$):

$$SP = H_{hydrate} \times A \times s_{hydrate} \times \Phi \times 164 \times \rho_{CH4_{atmospheric}}$$

The resulting seepage potential for the modelled area lies between 26 and 212 Mt.

**Data availability**. The data sets analysed during the current study are available from the corresponding author on reasonable request.

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

## Acknowledgements

The research leading to these results has received funding from the People Programme (Marie Curie Actions) of the European Union's Seventh Framework Programme FP7/2007-2013/ under REA grant agreement n° 317217. The research forms part of the GLANAM (GLAciated North Atlantic Margins) Initial Training Network. We would like to thank Anna Hughes from the University of Bergen for giving us access to her ice-sheet database and helping us creating the overview map in Fig. 1a. Furthermore, we would like to thank GEBCO to providing global bathymetric data by the British Oceanographic Data Centre used in Fig. 1a. We would like to thank the Norwegian Petroleum Directorate for providing 3D seismic data used in Fig. 3b and Norsk Hydro AS for providing the bathymetry data used in Fig. 6.We would like to thank the Department of Earth Science of the University of Bergen for providing funding for publishing this manuscript. J.M. is member of the Centre for Arctic Gas Hydrate, Environment and Climate (CAGE), which was supported by the Research Council of Norway through its Centres of Excellence funding scheme grant No. 223259. S.P. acknowledges the support from the Research Council of Norway through its Center of Excellence funding scheme, project 223272 (CEED).

## Author contributions

J.K., H.H., C.B., S.P. designed the study and wrote the manuscript. L.B. and J.K. compiled the sediment history reconstruction and J.K. and L.R. performed heat flow and gas hydrate stability simulations. V.L. and M.S. carried out the analysis of a sea floor sediment samples collected by J.M. H.H. collected and analysed sediment cores. All authors helped edit the manuscript.
