## [Peer review file · Nature Communications]

Reviewers' comments:

Reviewer #1 (Remarks to the Author):

Review of Karstens et al, Glacigenic sedimentation pulses triggered post-glacial gas hydrate dissociation

by W. Wood

Excellent topic, highlighting a new line of thinking in gas hydrate stability and slope stability on continental margins.

“Our simulations indicate that sedimentation on its own is able to cause significant hydrate dissociation independent of bottom water temperature warming or sea level fluctuation”

This is a very important and exciting result, and worthy of publication in this journal. I recommend publication after some minor, but not trivial edits. I think the authors can address all of my comments in a day or so of work. No new analyses are required, only perhaps a few simple calculations.

As I understand the manuscript, the main point is that methane seepage correlates with high sedimentation rates. I don't know enough about dolomite chemistry to know for sure, but I don't see how it adds to the story. Similarly, I don't see how photographs of the biology help the story. The story is the PT modeling, and the correlations (if not cause and effect) of glacial sedimentation, the shifting of the PT boundary and its effect on gas and gas hydrate accumulations.

I have a couple of scientific issues, regarding the buoyancy and pressure calculations. Specifically:

“the buoyancy of gas and related volumetric expansion of the pore fluid during hydrate dissociation will cause increasing pore overpressure and the formation or reactivation of focused fluid conduits involving the cracking of sediment formations. ”

The buoyancy of gas bubbles is generally not enough to overcome the surface tension. If the authors make the argument of buoyancy, they should move the pressure and density calculations to the main body of the text. Also, the pressure arguments used in these calculations I believe require the assumption of connectivity – the gas phase must be vertically connected throughout the sediment column. Is this realistic in these sediments? Otherwise the surface tension becomes a major impediment to the buoyancy.

The authors also argue that the dissociating gas hydrate increases the pressure. It seems to me that the sediment load itself likely puts far more stress on the system than changes to pressure from dissociation, and far more than buoyancy. I suspect that the dewatering from below the GHSZ is responsible for carrying the newly dissociated CH₄ to the seafloor, but that also would require some calculation, perhaps from the model?

In the end the authors have shown very clearly the correlation between rapid sedimentation and seafloor methane flux, but have not, in my opinion, clearly shown how the conduits have opened up (which is far more difficult). I think the quantitative modeling and analyses of the carbon in the biological samples showing the correlation of sedimentation rate to seafloor methane flux is, by itself very significant and worthy of publication.

Other comments:

“The temporal and spatial correlation between the maximum gas hydrate dissociation and the pockmark activity indicates that gas hydrate dynamics related to increased glacigenic sediment accumulation controlled methane seepage after the Last Glacial Maximum at the Nyegga pockmark field. ”

I would use the word “suggests” or “strongly suggests” instead of “indicates”.

“it is likely that the sudden hydrate dissociation due to sediment input has contributed to the

increase of global atmospheric methane concentrations after the Last Glacial Maximum³⁵ because the sudden overpressure build-up has formed new migration pathways. "

This reaches too far without more evidence. There is certainly consistency, but "likely" implies more certainty, and requires estimates of how the gas makes it through the water column, which is not obvious.

Minor comments

Abstract typo at "Maximum at for mid-Norwegian "

The authors should go through the text and figures carefully to make sure all the captions are consistent with the actual figures and with the body of the text.

Fig 1 caption

I don't see GB on the map (ok to eliminate it from discussion, it's not required)

Caption needs work. We don't need to see echo sounder data here, seismic is enough, but remove reference to Auth carbonates (Fig 1 c) in text.

Fig. 2 caption:

I would stress that Fig2 b is at a single location.

Fig 3 caption:

I don't see "phases with enhanced methane seepage. " in the figure

Need words for Fig 3g

Reviewer #2 (Remarks to the Author):

Summary

This is a very interesting paper and one of the first of its kind to closely study the effect of glacial sedimentation on the marine gas hydrate system stability during the last glaciation-early deglaciation period. The authors integrate a robust age-depth model with the stratigraphic framework in order to reconstruct the spatial sedimentation history of the Nyegga pockmark field. This 4D reconstruction is then combined with a heat flow model and empirical variation in sea level to reconstruct gas hydrate stability from 30 ka until 15 ka. The results of this transient simulation illustrate that gas hydrate would have mostly dissociated between 18 and 17 ka in response to an increase of glacial sediment input. This timing nicely matches with radiocarbon ages of authigenic carbonate crusts and published fossil chemosynthetic bivalves. The manuscript is generally clearly written and the arguments are well presented. I believe the manuscript presents a significant advance in the field and is worth publishing.

Main Comments

-Introduction

A major assumption made by the authors is the constant bottom water temperature (3rd paragraph). In fact, benthic $\delta^{18}O$ values see Dokken and Jansen (1999 Nature, ref. 14/fig.2) show a sharp decrease during HS1 event (~ 18-15 ka). Although the interpretation of this signal is matter of debate, it has been attributed to brine formation (Dokken and Jansen 1999 Nature), fresh meltwater (Standford et al. 2011 QSR) or warming of intermediate water masses (e.g. Ezat et al. 2014 Geology). The latter hypothesis implies that the assumption of constant bottom water temperature may not be valid, thus, I would suggest the authors to consider an alternative scenario assessing the impact of bottom water temperature warming.

-Results

Why do you assume that the (authigenic?) carbonate samples are somewhat related to methane seepage? Their high $\delta^{13}C$ values (from 0 to -5 permil) apparently preclude incorporation of methane-derived carbon, instead, the C-O isotope values are similar to bulk sediment measurements for the area (see ref. 26). Thus the origin of these carbonates could be questioned.

-Discussion

On p. 5, the authors state, "it is likely that the sudden hydrate dissociation due to sediment input has contributed to the increase of global atmospheric methane concentrations after the Last Glacial Maximum". This statement is highly speculative and oversells the paper. The postglacial increase in atmospheric methane concentration has been dominantly attributed to wetland emissions see for example Sowers et al. (2006, Science). Moreover, McGinnis et al. (2006, JGR) showed that bubbles emitted at water depths deeper than ~150 m are unlikely to reach the atmosphere.

Minor Comments

The abstract does not mention what methodology has been used. It should also mention the water depths of the studied area.

Paragraph 2/Fig. 2: References are missing. Also, it is not clear if this model (fig. 2) has been drawn in reference to other papers. If this is a working hypothesis than it should be clearly stated. The authors may not be aware of recent papers suggesting that glacial loading may have been controlling gas hydrates stability during the last glacial maximum. I think one of these following papers should be cited in this manuscript:

Andreassen, K., Hubbard, A., Winsborrow, M., Patton, H., Vadakkepuliambatta, S., Plaza-Faverola, A., ... & Mienert, J. (2017). Massive blow-out craters formed by hydrate-controlled methane expulsion from the Arctic seafloor. *Science*, 356(6341), 948-953.

Crémière, A., Lepland, A., Chand, S., Sahy, D., Condon, D. J., Noble, S. R., ... & Brunstad, H. (2016). Timescales of methane seepage on the Norwegian margin following collapse of the Scandinavian Ice Sheet. *Nature communications*, 7.

Portnov, A., Vadakkepuliambatta, S., Mienert, J., & Hubbard, A. (2016). Ice-sheet-driven methane storage and release in the Arctic. *Nature communications*, 7.

Winsborrow, M., Andreassen, K., Hubbard, A., Plaza-Faverola, A., Gudlaugsson, E., & Patton, H. (2016). Regulation of ice stream flow through subglacial formation of gas hydrates. *Nature Geoscience*, 9(5), 370-374.

Paragraph 3: Just a thought, is there anything known about a potential forebugle effect associated with glacial loading?

Fig. 4 could be relegated to supplements.

Supplements

Tab.1 Please clarify if 14C ages are corrected or not from reservoir effect. Both radiocarbon ages should be presented

Sensitivity analysis: more details are required about the different scenarios.

Reviewer #3 (Remarks to the Author):

The authors modelled hydrate decomposition at the base of a paleo-hydrate stability zone. They try to explain past seepage on the sea floor of southern Voering Plateau by pulses of hydrate dissociation, which creates pore-overpressure that trigger the focused fluid flow. Based on their model, they infer the highest gas hydrate decomposition at around 18 ka BP. They attribute the increased dissociation of hydrate to increased sedimentation rate, which leads to an uplift of the GHSZ. In general, the considerations are right and it is nice that an increase of gas hydrate decomposition is shown in the model, however, the proof of seepage on the sea floor is weak and

the statement of the paper should be made much more careful.

Two major problems occur:

1) The depth of the base of the ubiquitous blowout pipes in the area of the Nyegga pockmark field is not correlating with the depth of the postulated gas hydrate decomposition. As shown in Fig. 1c the chimneys start deeper, and therefore those structures must have formed much earlier. This was shown by Plaza-Faverola et al. (2011) in 3D-seismic survey-data, and interpreted by repeated events of focused fluid expulsion. In the paper itself the authors talk about a possible reactivation of conduits, however, in the abstract they generalize too much to the point that the blowout pipes are formed during the postulated event at 18 ka. This statement is therefore over-interpreted and the authors should be much more careful.

2) Radiocarbon dating of seep carbonates in general is very critical, because older carbon from the sediment is involved and at that complicate mineralogy (mixture of carbonate phases) is involved, as shown in the paper. U-Th dating of the carbonates would be much better!

The dating of the fossil shell of the bivalve is crucial as well, because of possible diagenic influence on the shell.

I therefore do not recommend the paper for publication in NATURE Communications.

Reviewer #1 (Remarks to the Author):

Excellent topic, highlighting a new line of thinking in gas hydrate stability and slope stability on continental margins.

“Our simulations indicate that sedimentation on its own is able to cause significant hydrate dissociation independent of bottom water temperature warming or sea level fluctuation”

This is a very important and exciting result, and worthy of publication in this journal. I recommend publication after some minor, but not trivial edits. I think the authors can address all of my comments in a day or so of work. No new analyses are required, only perhaps a few simple calculations.

As I understand the manuscript, the main point is that methane seepage correlates with high sedimentation rates. I don't know enough about dolomite chemistry to know for sure, but I don't see how it adds to the story. Similarly, I don't see how photographs of the biology help the story. The story is the PT modeling, and the correlations (if not cause and effect) of glacial sedimentation, the shifting of the PT boundary and its effect on gas and gas hydrate accumulations.

I have a couple of scientific issues, regarding the buoyancy and pressure calculations. Specifically:

- 1) “the buoyancy of gas and related volumetric expansion of the pore fluid during hydrate dissociation will cause increasing pore overpressure and the formation or reactivation of focused fluid conduits involving the cracking of sediment formations. “ The buoyancy of gas bubbles is generally not enough to overcome the surface tension. If the authors make the argument of buoyancy, they should move the pressure and density calculations to the main body of the text.*

We moved the pressure calculation to the main text in lines 145 – 154: **“Excess pore pressure quantification. The overpressure resulting from the buoyancy of free gas can be calculated by $P = H_{\text{gas}} * (g * (\rho_w - \rho_{\text{CH}_4, 10 \text{ MPa}}))$ (ref. 35), where H_{gas} represents the gas column height, g is gravitational acceleration (9.81 m/s^2), ρ_w is density of the formation water ($\sim 1025 \text{ kg/m}^3$) and $\rho_{\text{CH}_4, 10 \text{ MPa}}$ is density of methane for pressure of 10 MPa C ($\sim 84 \text{ kg/m}^3$; ref. 36). The volume of the gas formed by gas hydrate dissociation is 164 times greater than the hydrate volume under atmospheric pressure conditions³⁷, while density of methane under atmospheric pressure condition (0.656 kg/m^3) is about 128 times smaller than at 10 MPa. The gas column height is consequently: $H_{\text{gas}} = 164/128 H_{\text{hydrate}} * S_{\text{hydrate}}$, where S_{hydrate} is the gas hydrate concentration (3 – 12%, ref. 38) and H_{hydrate} is the height of dissociated gas hydrates (40 m from simulations). The resulting overpressure due to buoyancy lies between 15.1 and 60.6 kPa.”**

- 2) Also, the pressure arguments used in these calculations I believe require the assumption of connectivity – the gas phase must be vertically connected throughout the sediment column. Is this realistic in these sediments? Otherwise the surface tension becomes a major impediment to the buoyancy. The authors also argue that the dissociating gas hydrate increases the pressure. It seems to me that the sediment load itself likely puts far more stress on the system than changes to pressure from dissociation, and far more than buoyancy. I suspect that the dewatering from below the GHSZ is responsible for carrying the newly dissociated CH4 to the seafloor, but that also would require some calculation, perhaps from the model?*

We agree with the reviewer that our calculations assume mobile, connected gas. A good indicator for these assumptions is the pronounced bottom-simulating reflector, which indicates the presence of significant amounts of free gas beneath the BGHSZ. Our simulations indicate a very dynamic change of the BGHSZ with time, while the sediments above the BGHSZ have a high gas hydrate concentration (3 – 12%). Such hydrate concentrations are unlikely to have formed within the glacial sediments by in-situ microbial processes and can only be explained by the upward migration of methane of methane due to hydrate recycling at the BGHSZ. Therefore, we believe that the assumption of a high gas mobility through the sediment column is plausible. The IODP drilling at Blake Ridge has revealed very high pore pressure beneath a BSR (Hornbach et al., 2004: “Critically pressured free-gas reservoirs below gas-hydrate provinces”). This observation is in good agreement

with vertical gas migration due to hydrate recycling and excess pore pressure build-up suggesting that this assumption is viable

To clarify this assumption, we have added a sentence highlighting the uncertainty to the discussion in lines 169 - 171: **“This calculation assumes that gas formed by the gas hydrate dissociation is mobile and accumulates beneath the BGHSZ, which is plausible considering the well-developed bottom-simulating reflector indicating the presence of a gas column beneath the highly dynamic BGHSZ (Fig. 3).”**

- 3) *In the end the authors have shown very clearly the correlation between rapid sedimentation and seafloor methane flux, but have not, in my opinion, clearly shown how the conduits have opened up (which is far more difficult). I think the quantitative modeling and analyses of the carbon in the biological samples showing the correlation of sedimentation rate to seafloor methane flux is, by itself very significant and worthy of publication.*

The opening of the conduit is very difficult to proof and there are only few sandbox models or numerical simulation based studies dealing with the opening or formation of focused fluid conduit. Our simulations cannot deliver any input to this issue, but our simulations predict indications that this process occurred at Nyegga during the LGM, which is agreement with the seismic data showing that the pipe structures crosscut LGM sediments (new figure 2, old figure 1c). Additionally, the 3D seismic analysis by Plaza-Faverola et al. (2011), indicates an episodic glaciation controlled activity of the pipes. We cannot rule out that the conduits have been open before 18,000 BP. However, the fact that chemosynthetic bivalves have not been found in deeper sediment layers of the last glaciation is in our opinion a strong indicator for strongly enhanced methane transport during the LGM.

Other comments:

- 4) *“The temporal and spatial correlation between the maximum gas hydrate dissociation and the pockmark activity indicates that gas hydrate dynamics related to increased glacial sediment accumulation controlled methane seepage after the Last Glacial Maximum at the Nyegga pockmark field. “ I would use the word “suggests” or “strongly suggests” instead of “indicates”.*

We changed “indicate” to **“strongly suggest”** in line 163.

- 5) *“it is likely that the sudden hydrate dissociation due to sediment input has contributed to the increase of global atmospheric methane concentrations after the Last Glacial Maximum³⁵ because the sudden overpressure build-up has formed new migration pathways. “ This reaches too far without more evidence. There is certainly consistency, but “likely” implies more certainty, and requires estimates of how the gas makes it through the water column, which is not obvious.*

We changed the section dealing with this issue (see comment 17).

Minor comments

- 6) *Abstract typo at “Maximum at for mid-Norwegian “*

We changed “for” to **“the”** in line 19.

- 7) *The authors should go through the text and figures carefully to make sure all the captions are consistent with the actual figures and with the body of the text.*

We followed this advice.

- 8) *Fig 1 caption: I don’t see GB on the map (ok to eliminate it from discussion, it’s not required)*

GS on the map should be GB. We changed this.

- 9) *Cation needs work. We don't need to see echo sounder data here, seismic is enough, but remove reference to Auth carbonates (Fig 1 c) in text.*

We removed it.

- 10) *Fig. 2 caption: I would stress that Fig2 b is at a single location.*

We have added “**for a single location**” in line 254 (new figure 3).

- 11) *Fig 3 caption: I don't see “**phases with enhanced methane seepage.**” in the figure*

We refer to the vertical lines in subfigures e) and f).

- 12) *Need words for Fig 3g*

The captions for f) and g) got lost during editing. We included f) in lines 266:” **f) Gas hydrate dissociation rate and phase with enhanced methane seepage.**” 3g) is now 5a) “**Map showing the distribution of gas hydrate dissociation at 17.8 ka with pockmark location9, extent of gas hydrates10 (dashed black lines) and coring and sample locations.**” In lines 271 – 272.

Reviewer #2 (Remarks to the Author):

Summary

This is a very interesting paper and one of the first of its kind to closely study the effect of glacial sedimentation on the marine gas hydrate system stability during the last glaciation-early deglaciation period. The authors integrate a robust age-depth model with the stratigraphic framework in order to reconstruct the spatial sedimentation history of the Nyegga pockmark field. This 4D reconstruction is then combined with a heat flow model and empirical variation in sea level to reconstruct gas hydrate stability from 30 ka until 15 ka. The results of this transient simulation illustrate that gas hydrate would have mostly dissociated between 18 and 17 ka in response to an increase of glacial sediment input. This timing nicely matches with radiocarbon ages of authigenic carbonate crusts and published fossil chemosynthetic bivalves. The manuscript is generally clearly written and the arguments are well presented. I believe the manuscript presents a significant advance in the field and is worth publishing.

Main Comments

- 13) *Introduction: A major assumption made by the authors is the constant bottom water temperature (3rd paragraph). In fact, benthic $d_{18}O$ values see Dokken and Jansen (1999 Nature, ref. 14/fig.2) show a sharp decrease during HSI event (~ 18-15 ka). Although the interpretation of this signal is matter of debate, it has been attributed to brine formation (Dokken and Jansen 1999 Nature), fresh meltwater (Standford et al. 2011 QSR) or warming of intermediate water masses (e.g. Ezat et al. 2014 Geology). The latter hypothesis implies that the assumption of constant bottom water temperature may not be valid, thus, I would suggest the authors to consider an alternative scenario assessing the impact of bottom water temperature warming.*

We strongly agree with reviewer. The evolution of the bottom water temperatures at the mid-Norwegian margin during that period is a complex and controversial topic. We actually tested several scenarios and show the results in the sensitivity analysis in the method section. In figure A.3a) we show the results for our simulation using different temperature curves and all result in similar gas hydrate dissociation rate and absolute gas hydrate dissociation curves. However, the tested scenarios are, as indicated in comment 24, only poorly explained. We now highlight the controversy in text in lines 74 –76 “**The stable oxygen isotope analysis of benthic foraminifera reveals a pronounced $d_{18}O$ anomaly around the LGM, which may be explained by brine formation16, freshening by melt water input17 or warming of immediate water masses¹⁸”**

Furthermore, we refer to the sensitivity analysis in lines 84 –89:” **We conducted a sensitivity analysis for the input parameters of our simulations by testing different bottom water temperature profiles, sedimentation rate reconstructions, subsidence rates, thermal diffusivity values and geothermal gradients. The sensitivity analysis revealed that variations of these input parameters had no significant impact on the timing and trend of gas hydrate dissociation (see Methods Fig A.3). This includes simulation scenarios, which assume a bottom water warming before or during the LGM.**

Finally, we have updated the description of the sensitivity analysis (see comment 24).

14) Results: Why do you assume that the (authigenic?) carbonate samples are somewhat related to methane seepage? Their high $d^{13}C$ values (from 0 to -5 permill) apparently preclude incorporation of methane-derived carbon, instead, the C-O isotope values are similar to bulk sediment measurements for the area (see ref. 26). Thus the origin of these carbonates could be questioned.

We agree that our measured $d^{13}C$ -data of bulk sediments containing MgCa-carbonate do not represent a ^{12}C -enriched endmember carbonate precipitated due to methane oxidation. However, the bulk $d^{13}C$ -values are clearly more negative than background data (+0.4 permill, SD: 0.2 permill) published for the working area (Paull et al., 2008). Moreover, our calculated $d^{13}C = -11$ permill of dolomite plots on a mixing line between background and pockmark/chimney carbonates (Fig. 4 in Paull et al., 2008). Thus, we can argue that biogenic carbonate was partly recrystallized during diagenesis and pore water is cemented due to microbially induced carbonate precipitation. Both processes are probably partly affected by incorporation of ^{12}C -rich carbon from an additional source. The ^{12}C -rich (deep or shallow) source is ambiguous and could be microbial methane, thermogenic methane, and/or higher hydrocarbons. In any case, the data points to a ^{12}C -rich carbon source, which can be related to pockmark activity as stated by other authors for this area. We would like to keep the section dealing with this topic. The text already mentions the uncertainty and that the measured values represent an endmember mixture.

15) Discussion: On p. 5, the authors state, "it is likely that the sudden hydrate dissociation due to sediment input has contributed to the increase of global atmospheric methane concentrations after the Last Glacial Maximum". This statement is highly speculative and oversells the paper. The postglacial increase in atmospheric methane concentration has been dominantly attributed to wetland emissions see for example Sowers et al. (2006, Science). Moreover, McGinnis et al. (2006, JGR) showed that bubbles emitted at water depths deeper than ~150 m are unlikely to reach the atmosphere.

We agree that most methane from natural seeps does not reach the atmosphere via direct bubble transport. However, the paper by McGinnis et al., 2006 also states that “Therefore only a catastrophic bubble release will contribute significantly to the direct methane transport from deep water seeps (>100 m) to the atmosphere”. We believe that catastrophic, blowout-like events represent one endmember of plausible scenarios for the creation or reactivation of the pipe structure. Nevertheless, we agree that our previous statement is too speculative and we have highlighted this in the manuscript in lines 183 – 196:

“Even if a large fraction of the methane has been recycled within the hydrate stability zone or consumed by the benthic filter, it is likely that the sudden hydrate dissociation due to sediment input has released large amounts of methane into the water column. It is difficult to constrain, if the methane release from pockmarks at Nyegga occurred continuously with low seepage rates or catastrophically with high seepage rates comparable to drilling-induced blowout events. Numerical simulations indicate that methane released from natural seeps deeper than 100 m below sea level will not reach the atmosphere via bubble transport due to oxidation and dissolution, while a catastrophic methane release allows a more efficient transport of methane to

the sea surface⁴². Consequently, it is speculative if the focused methane release at Nyegga after the LGM directly affected the atmospheric methane budget. Nevertheless, the dissolution and oxidization of methane in the water column has likely reduced the ocean's potential of absorbing atmospheric methane. Assuming that the sedimentation controlled gas hydrate dissociation and methane venting occurred simultaneously for different glaciated gas hydrate provinces after the LGM, this process would have contributed to the increase of global atmospheric methane concentrations after the Last Glacial Maximum⁴³.”

Minor Comments

16) *The abstract does not mention what methodology has been used. It should also mention the water depths of the studied area.*

We added information about the methodology in line 17 and the water depth in lines 20 – 21.

17) *Paragraph 2/Fig. 2: References are missing. Also, it is not clear if this model (fig. 2) has been drawn in reference to other papers. If this is a working hypothesis than it should be clearly stated.*

The presented process of gas hydrate dissociation due to sedimentation is one aspect of the well-established concept of gas hydrate recycling, which describes the redistribution of gas hydrates within the sediment column by different external forces including sedimentation, tectonic uplift, sea-level change or bottom-water-warming. There are several case studies and numerical simulations based studies, which include the effect of sedimentation on gas hydrate systems (e.g. Kvenvolden, 1993; Haacke et al., 2007; Pecher et al., 1996; Xu et al., 2006; Riedel et al., 2011, etc.). At least to our knowledge, sedimentation is only minor aspect in these studies and none of these studies explains the general concept as detailed as we do. Therefore, we believe that it would neither be adequate to cite one of these papers in the figure captions, nor to claim that our concept is new. We added two sentences in lines 44 – 50 including two new references to the general concept of gas hydrate recycling:

“The dynamic redistribution of gas hydrates within the sediment column due to external forces is a well-established process known as gas hydrate recycling^{12, 13}. During glacial cycles, gas hydrate dynamics is governed by sea level change, regional bottom water temperature fluctuations and local sedimentation rate changes. The local sea floor depth is controlled by the global sea level and regional uplift or subsidence. Bottom water temperature changes during deglaciation are mainly controlled by the reorganization of ocean currents and warming of water masses affecting the sediment temperature profile. This study has a focus on the impact of sedimentation on gas hydrate dynamics.”

18) *The authors may not be aware of recent papers suggesting that glacial loading may have been controlling gas hydrates stability during the last glacial maximum. I think one of these following papers should be cited in this manuscript: Andreassen et al., 2017; Cremiere et al., 2016; Portnov et al., 2016; Winsborrow et al., 2016*

We are aware of the publications, which deal with the impact of ice sheet loading on gas hydrate controlled processes. In case of Nyegga, glacial loading had no influence on gas hydrate dynamics during the LGM, because the study area is located in water depth greater than 600 m and was not covered by grounded ice.

19) *Paragraph 3: Just a thought, is there anything known about a potential forebugle effect associated with glacial loading?*

To our knowledge, there is no published information about such an effect for the study area. Therefore, we only use the local subsidence trend published by Dahlgreen et al., (2002).

20) Fig. 4 could be relegated to supplements.

We now combined it with the previous subfigure 3g) into the new figure 5. Now it is not receiving so much focus as before.

21) *Supplements: Tab.1 Please clarify if ^{14}C ages are corrected or not from reservoir effect. Both radiocarbon ages should be presented*

These dates are not corrected. As mentioned in the paragraph about radiocarbon age calibration, we used a local reservoir effect of 400 years with an uncertainty of 200 years. Therefore, we would need to include three additional dates (minimum, maximum, mean) into the table. We think that this would not add valuable information, but rather distract the reader. However, we have added the information that the radiocarbon dates are corrected to the table caption in line 458 **“Radiocarbon ages are not corrected from reservoir effect”**. In addition, we added a column giving the references to the radiocarbon dates.

22) *Sensitivity analysis: more details are required about the different scenarios.*

We have updated the sensitivity analysis description in lines 548 - 574:

“Sensitivity analysis. In order to test the robustness of our simulations, we performed sensitivity analyses for different sedimentation rates, bottom water temperature profiles, the subsidence rates affecting the local sea level curve, bulk thermal conductivity values and the geothermal gradients (Fig. A.3; the input parameter and simulations results used in the main part of the manuscript are marked with dashed black lines). We tested the impact of bottom water temperature on the gas hydrate dynamics by applying different temperature evolution profiles. These simulations included scenarios with constant temperatures of -1°C (dashed black line in Fig. A.3a) and -2°C (yellow line) and repeatedly fluctuating temperatures between -1.75 and -2.25°C (orange line). These input parameters result in very similar curves for the dissociation rate of gas hydrates. In addition, we performed simulations with four potential warming scenarios with an absolute warming of $\sim 0.5^{\circ}\text{C}$ (green line) and $\sim 1^{\circ}\text{C}$ (red, light blue and dark blue). All simulations result in similar shaped gas hydrate dissociation curves with a peak of dissociation after 18,000 years before present. The absolute thickness of gas hydrate dissociation at the end of the simulations lies between 35 and 47 m, which shows that bottom water temperatures had only a secondary impact on gas hydrate dynamic during the LGM at Nyegga. The solution space of the Bayesian age-depth reconstruction allow different sedimentation rate reconstructions (Fig. A.3b). In addition to the statistically most likely reconstruction (dashed black line), we tested the sedimentation rate reconstructions, which bound the solution space (blue and yellow lines; marked in grey in Fig. A.2f). These sedimentation rate reconstructions result in very similar gas hydrate dissociation rates and absolute dissociation curves. The analysis of different local subsidence rate (Fig. A.3c), which influence the local sea level curve reveals that subsidence had only a minor impact on gas hydrate dynamics, even when neglecting subsidence (yellow line) or assuming a twice as high subsidence rate (blue line). The analyses of the thermal diffusivity of the sediments and the geothermal gradients reveal that the effect of sedimentation on gas high dynamics becomes more important for sediments with a high thermal diffusivity and in areas with high geothermal gradients (Fig. A.3d and e). However, the timing of enhanced gas hydrate dissociation remains stable for various thermal property values. The sensitivity analysis revealed that the gas hydrate dissociation rate and the absolute value of dissociated gas hydrates are robust and similar gas hydrate dynamics can be observed for a wide range of parameters.”

Reviewer #3 (Remarks to the Author):

The authors modelled hydrate decomposition at the base of a paleo-hydrate stability zone. They try to explain past seepage on the sea floor of southern Voering Plateau by pulses of hydrate dissociation, which creates pore-overpressure that trigger the focused fluid flow. Based on their model, they infer

the highest gas hydrate decomposition at around 18 ka BP. They attribute the increased dissociation of hydrate to increased sedimentation rate, which leads to an uplift of the GHSZ.

- 23) *In general, the considerations are right and it is nice that an increase of gas hydrate decomposition is shown in the model, however, the proof of seepage on the sea floor is weak and the statement of the paper should be made much more careful.*

The focused fluid conduits at Nyegga crosscut sediments deposited before and during the LGM. This observation alone is already a clear indicator for seepage at the seafloor at the LGM or later. We therefore disagree with the general statement that the proof for seepage at the seafloor is weak. All further age constraints are only aiming to pinpoint the timing of seepage. However, we have clarified this in the manuscript by adding three sentences in lines 120 – 124:

”Reflection seismic data and radiocarbon dating of sediment cores reveal that the pipes crosscut sediments, which were deposited before and during the LGM (Fig. 2). This is a clear indicator for seepage at the end of the LGM or afterwards. To constrain the timing of enhanced methane seepage during that period further, we analysed seep fauna and seafloor carbonates.”

Two major problems occur:

- 24) *The depth of the base of the ubiquitous blowout pipes in the area of the Nyegga pockmark field is not correlating with the depth of the postulated gas hydrate decomposition. As shown in Fig. 1c the chimneys start deeper, and therefore those structures must have formed much earlier. This was shown by Plaza-Faverola et al. (2011) in 3D-seismic survey-data, and interpreted by repeated events of focused fluid expulsion. In the paper itself the authors talk about a possible reactivation of conduits, however, in the abstract they generalize too much to the point that the blowout pipes are formed during the postulated event at 18 ka. This statement is therefore over-interpreted and the authors should be much more careful.*

The reviewer is correct in pointing out that the pipes may have existed before the major degassing episode (as we have discussed in the text). We have clarified this point by adding “**or reactivation**” to line 22 in the abstract.

- 25) *Radiocarbon dating of seep carbonates in general is very critical, because older carbon from the sediment is involved and at that complicate mineralogy (mixture of carbonate phases) is involved, as shown in the paper. U-Th dating of the carbonates would be much better! The dating of the fossil shell of the bivalve is crucial as well, because of possible diagenic influence on the shell.*

We agree that the potential presence of “older carbon” makes the interpretation of the carbonates complicated. Already in the first version of the manuscript we had stated that the radiocarbon dates of the carbonate have to be understood as maximum formation times (lines 132 and 512 of the revised manuscript).

On first glance U-Th dating is the method of choice for constraining the age of seep carbonates. Because of this we have applied different U-Th dating methods (e.g. Bayon et al., 2009; Liebetrau et al. 2014). However, it turned out that the U-Th analyses were not successful and cannot provide reliable isochrones. We analyzed three to four subsamples of three different carbonate samples from the study area including the ones presented in table A.2 (Mie1-3729-1 to -3). We also leached mechanically separated dolomitic and calcitic parts comparing the results with the homogeneous bulk of the main sample in order to achieve an isochron spread as large as possible. However, none of the datasets provided a reasonable isochron and they suggested different formation ages for the calcitic and dolomitic carbonates, with the latter being the younger fraction. Therefore, U-Th dating could not contribute to constraining the formation age. Colleagues in Bergen had similar problems and were not successful with U-Th dating on Nyegga carbonate samples (personal communication).

We decided not including the U-Th discussion in the manuscript, because a detailed discussion of the U-Th data would only contribute to a better understanding of U-decay series systematics in this fine-

grained poly-phase carbonates but distract from the actual main topic of the manuscript. Therefore, we decided to rely on the less precise but more robust assumption of ^{14}C data as maximum formation ages. We added a statement about this topic in line 512 - 519:

“In addition, we applied U-Th geochronology methods for methane-derived authigenic carbonates to constrain the precipitation age of the carbonates^{53,54}. However, the analysis indicated different formation ages for the calcitic and dolomitic dominated subsamples of the same carbonate sample and did not provide any reasonable isochron-based age information. This result is in agreement with the light stable isotope findings. The ^{14}C data from the carbonate crust sample Mie 1-3729-1 represent a rather robust maximum formation age of 18,000 years BP (15,710 ^{14}C). This maximum age interpretation is based on the assumption that any carbon contributed from the sediment system below would be relatively depleted in ^{14}C by ongoing decay during burial, resulting in apparent higher ages.”

We agree that the dating of fossil shells is important and we have been citing the two papers (Paull, et al., 2008 and Chen et al., 2011) already in the original submission.

REVIEWERS' COMMENTS:

Reviewer #1 (Remarks to the Author):

Remarks to the Author:

Review of Karstens et al, Glacigenic sedimentation pulses triggered post-glacial gas hydrate dissociation

by W. Wood

The authors have more than satisfactorily addressed my concerns, and in my opinion the article should be published as is.

Changing the figures helped significantly, they seem to support the text much better. I think much of my original confusion was alleviated by the authors careful attention to aligning the text, captions, and figures. Also, the modeling is more much clearly presented.

I'm still not convinced that a photo of the bivalves is necessary, but it's role in the story makes more sense now.

Reviewer #2 (Remarks to the Author):

I have read the revised version of the manuscript.

The authors have done a nice job in addressing most of the reviewers' comments, therefore, I consider that the manuscript is almost in shape for being accepted. However, I would suggest few minor corrections;

- l.47 to 51 would benefit from few references

- l.75 change "d18O" to $\delta^{18}\text{O}$

-l.520 to 524 do not provide any useful information. What does "reasonable isochron-based age information" mean? Is-it because detrital Th contribution is too high to provide any reliable age?

Reviewer #4 (Remarks to the Author):

The study by Karstens et al. is an excellent piece of work and the mechanism that trigger gas hydrate dissociation presented here has a high potential for understanding the dynamics of hydrate and its link to sediment loading and slope stability during glacial-interglacial cycles. The manuscript is definitely suitable for publication in Nature Communications, novel and of broad relevance and interest for the geosciences community.

I think the reviewer 3's concerns have been adequately addressed by the authors. I have a few additional minors for the authors to consider during next round of revision.

Specific comments on the manuscript:

The terms 'seep' and 'vent' as well as 'seepage' and 'venting' are used interchangeably in this manuscript. I suggest sticking to 'seep' and 'seepage' unless the definition of the respective terms is improved.

Line 23-24: consider removing 'authigenic carbonate crust'. As mentioned by the authors, radiocarbon dating of authigenic carbonate represent the maximum age of methane seepage because the effect of fossil carbon. It is therefore critical to include what kind of samples (shell materials cemented by the carbonates, carbonate itself, or Foraminifera shells in the sediment) have been used obtaining radiocarbon dating (Tab. A.1).

Lines 19, 89, 138: better add an object after 'this'.

Lines 42, 69: change 'Fig.' to 'Figs.'.

Line 195: change 'oxidization' to 'oxidisation', be consistent throughout the manuscript.

Revision notes for “Glacigenic sedimentation pulses triggered post-glacial gas hydrate dissociation”

Reviewer #1 (Remarks to the Author):

The authors have more than satisfactorily addressed my concerns, and in my opinion the article should be published as is. Changing the figures helped significantly, they seem to support the text much better. I think much of my original confusion was alleviated by the authors careful attention to aligning the text, captions, and figures. Also, the modeling is more much clearly presented.

1) I'm still not convinced that a photo of the bivalves is necessary, but it's role in the story makes more sense now.

We agree with the reviewer that these photographs are not absolutely essential, but as there is enough space we would like to keep the photographs in the main part of the paper. For biologists the photographs show that this is seep fauna and corroborate the scientific argument.

Reviewer #2 (Remarks to the Author):

I have read the revised version of the manuscript.

The authors have done a nice job in addressing most of the reviewers' comments, therefore, I consider that the manuscript is almost in shape for being accepted. However, I would suggest few minor corrections:

2) l.47 to 51 would benefit from few references

We have added a reference to Bauch et al., 2001: A multiproxy reconstruction of the evolution of deep and surface waters in the subarctic Nordic seas over the last 30,000 yr.

3) l.75 change "d18O" to $\delta^{18}\text{O}$

We have changed it.

4) l.520 to 524 do not provide any useful information. What does "reasonable isochron-based age information" mean? Is-it because detrital Th contribution is too high to provide any reliable age?

This is exactly the point. A too high detrital Th contribution was the main problem for the of U-Th analysis. In more detail: Both, the calcite and dolomite dominated phases plot as straight ($R^2=0.99$) mixing lines when plotting the $^{230}\text{Th}/^{232}\text{Th}$ against the $^{238}\text{U}/^{232}\text{Th}$ activity ratio isochrons. However, these two lines have inverse slopes, which indicates that the two main carbonate phases have not formed at the same time (see also manuscript). Furthermore, the mean bulk of the samples and the dolomite-dominated subsamples are plotting very close to each other at extreme low $^{230}\text{Th}/^{232}\text{Th}$ activity ratios of 1.27 ± 0.01 and 1.18 ± 0.01 , respectively, and almost similar $^{238}\text{U}/^{232}\text{Th}$ activity ratios of 6.18 ± 0.05 and 6.4 ± 0.05 . These values are close to the typical detrital range for $^{230}\text{Th}/^{232}\text{Th}$ activity ratios (0.6 to 1.1). The calcite-dominated subsample reveals a slightly different signature (1.67 ± 0.02 ($^{230}\text{Th}/^{232}\text{Th}$) and 3.58 ± 0.03 ($^{238}\text{U}/^{232}\text{Th}$)), but could still not provide an accurate absolute age.

We decided to avoid discussing this in detail, because it would provide neither information relevant for the main topic of the manuscript nor important information for U-Th dating in general. Nevertheless, we added an additional statement in the manuscript indicating the issue of too high detrital Th contribution in lines 307-309: "A too high contribution of detrital Th in both carbonate

phases prevented deducing reasonable isochron-based age information. The finding of different formation ages for the different carbonate phases is in agreement with the light stable isotope findings.”

Reviewer #4 (Remarks to the Author):

The study by Karstens et al. is an excellent piece of work and the mechanism that trigger gas hydrate dissociation presented here has a high potential for understanding the dynamics of hydrate and its link to sediment loading and slope stability during glacial-interglacial cycles. The manuscript is definitely suitable for publication in Nature Communications, novel and of broad relevance and interest for the geosciences community. I think the reviewer 3's concerns have been adequately addressed by the authors. I have a few additional minors for the authors to consider during next round of revision.

Specific comments on the manuscript:

5) The terms 'seep' and 'vent' as well as 'seepage' and 'venting' are used interchangeably in this manuscript. I suggest sticking to 'seep' and 'seepage' unless the definition of the respective terms is improved.

We followed the advice of the reviewer and changed vent into seep in line 42 and venting into seepage in lines 143 and 198.

6) Line 23-24: consider removing 'authigenic carbonate crust'. As mentioned by the authors, radiocarbon dating of authigenic carbonate represent the maximum age of methane seepage because the effect of fossil carbon. It is therefore critical to include what kind of samples (shell materials cemented by the carbonates, carbonate itself, or Foraminifera shells in the sediment) have been used obtaining radiocarbon dating (Tab. A.1).

We agree with the reviewer that the authigenic carbonate crusts are a weaker age constrain than the bivalves. We removed mentioning of the authigenic carbonate crust from the abstract.

7) - Lines 19, 89, 138: better add an object after 'this'.

We have changed "this implies" into "Sedimentation pulses triggered" in line 20, "this" into "The sensitivity analysis" in line 91 and added "value" in line 140.

8) Lines 42, 69: change 'Fig.' to 'Figs.'

We have changed it.

9) Line 277: change 'oxidization' to 'oxidisation', be consistent throughout the manuscript.

We have changed it.